# Effect of *Buddleja cordata* Leaf Extract on Diabetic Nephropathy in Rats

**DOI:** 10.3390/ijms252111432

**Published:** 2024-10-24

**Authors:** Elizabeth Alejandrina Guzmán Hernández, Adriana Miranda Ocaña, Omar Ortiz Pedraza, María Eugenia Garín Aguilar, Rubén San Miguel Chávez, Martín Palomar Morales, David Segura Cobos

**Affiliations:** 1Medical Surgeon Career, Faculty of Higher Studies Iztacala, National Autonomous University of Mexico, Tlalnepantla 54090, State of Mexico, Mexico; seguracd@unam.mx; 2Biology Career, Faculty of Higher Studies Iztacala, National Autonomous University of Mexico, Tlalnepantla 54090, State of Mexico, Mexicopedraza_234@iztacala.unam.mx (O.O.P.); 3Pharmacobiology Laboratory, Biology Career, FES Iztacala, Universidad Nacional Autónoma de México, Tlalnepantla 54090, State of Mexico, Mexico; maragarin@yahoo.com; 4Montecillo Campus, Postgraduate College, Texcoco 56230, State of Mexico, Mexico; sami89@gmail.com; 5Diabetes Mellitus Metabolism Laboratory, Biology Career, Faculty of Higher Studies Iztacala, National Autonomous University of Mexico, Tlalnepantla 54090, State of Mexico, Mexico; martinpalomarmorales@gmail.com

**Keywords:** *Buddleja cordata*, diabetic nephropathy, NFκΒ

## Abstract

One of complications of diabetes mellitus is diabetic nephropathy. In Mexico, in traditional medicine, *Buddleja cordata* Humb. Bonpl. & Kunth, (“tepozán blanco”) is a shrub plant used for the treatment of rheumatic diseases, postpartum bath, stomachache, skin burns, diarrhea in children, sores, and cancer. Objectives: We examined the effect of methanol extract of leaves of *B. cordata* on the expression of pro-inflammatory cytokines and its antioxidant effects in diabetic nephropathy. Methods: We used the streptozotocin-induced diabetes mellitus model in rats; these were treated with methanol extract from leaves of *B. cordata* at 50 and 100 mg/kg (orally) for 4 weeks. Kidney weight/total body weight ratio and proteins/DNA, proteinuria and creatinine clearance, Western blot of nuclear factor κΒ (NFkB) p65 (cytoplasm and nucleus), peroxisome proliferator activated receptor gamma (PPARγ), and activities of glutathione peroxidase, superoxide dismutase, and catalase were determined.

## 1. Introduction

One of the complications of diabetes mellitus is diabetic nephropathy (DN). Although various mechanisms have been described that contribute to its development, the inflammatory process and fibrosis seem to play crucial roles. DN indicates chronic damage to kidney function, albuminuria, hypertension, tubulointerstitial fibrosis, a decrease in the glomerular filtration rate, and the thickening of the glomerular basement membrane [1,2]. There are various treatments used for DN, including hypoglycemic, angiotensin-converting enzyme inhibitors, angiotensin receptor blockers, and statins, or a regimen with a better lifestyle. Despite the effectiveness of these drugs, the appearance of side effects leads patients with ND to look for other treatment alternatives with fewer side effects, that are low cost, and that are within their reach, so they resort to medicinal plants accordingly. According to the World Health Organization, more than 80% of patients use medicinal plants at some point in the evolution of their disease [3].

In the search for new drugs in medicinal plants, some phytochemical compounds that possess more than one mechanism have been found, which will be useful for treating type 2 diabetes mellitus. For example, protocatechuic acid is an insulin sensitizer and secretagogue. Quercetin is an insulin sensitizer and secretagogue, as well as an inhibitor of a-glucosidase. Epicatechin inhibits a-glucosidase and is an insulin sensitizer. Kaempferol inhibits hepatic gluconeogenesis, is an insulin sensitizer, and inhibits a-glucosidase. Cafeic acid stimulates insulin secretion, reduces hepatic glucose output and a-glucosidase activity, and is an insulin sensitizer. 2-Hydroxy-destigloyl-6-deoxyswietenine acetate stimulates insulin secretion by a partial blockade of K+-ATP channels; a serotonergic modulation on 5-HT2 receptors inhibited hepatic glucose-6-phosphatase and increased muscle glucose uptake and basal oxygen consumption in muscle cells [4,5].

In Mexico, *Buddleja cordata* Humb. Bonpl. & Kunth, (“tepozán blanco”), zompantle; State of Mexico: rannazha (Otomi); Puebla: chkapungut; kaneje kuxindaa, kanda ku (Popoloca) belongs to the *Buddleja* genus, which comprises 125 species distributed worldwide, with the majority of species being reported in America; this continent has four centers of diversity, where Mexico is outstanding in harboring 20% of this diversity. Varieties including *B. americana L, B. parviflora Kunth, B. scordioides Kunth*, and *B. cordata* are widely distributed in Mexico and can be located from the state of Chihuahua to Yucatan. In traditional medicine, they are used in the form of a poultice to relieve wounds, ulcers, abscesses, and pimples. The Tarahumara ethnic groups use the leaves and bark to heal wounds and inflammation and a decoction of the root, leaves, and bark as a diuretic. Several studies have focused on demonstrating the biological properties of *B. cordata*, including its analgesic, anti-inflammatory, antibacterial, amebicide, antiparasitic, photoprotective, neuroprotective, antifungal, and embryotoxic effects. These effects have been associated with the presence of secondary metabolites such as verbascoside, luteolin, buddlejoside, iridioides, and acteoside [6,7,8,9].

Objectives: We examined the effect of methanol extract of leaves of *B. cordata* on the expression of pro-inflammatory cytokines and its antioxidant effect in diabetic nephropathy in streptozotocin-induced diabetic rats.

## 2. Results

An HPLC analysis of EMB revealed that the methanolic extract contains some phenolic acids, such as chlorogenic and ferulic. Flavonoids are represented in this species by catechin, phloridzin, naringenin, quercetin, and apigenin. Some terpenoids were detected such as acid ursolic, oleanolic, α-amyrin, and β-sitosterol. 

Of these compounds, the presence of α-amyrin stands out as the most abundant, with a concentration of 223.22, followed by ferulic acid at 56.74, oleanolic acid at 12.69, and catechin at 3.4 mg dry extract g^−1^ (Table 1 and Table 2 and Figure 1 and Figure 2).

### 2.1. In Vitro and In Vivo Antioxidant Activity

The methanolic extract of *B. cordata* was 30.12%; it possessed high phenolic contents (157 mg GAE/g of extract), while other plant that showed relevant antioxidant and medicinal properties, like *Chiranthodendron pentadacylon Larreat* (221.4 mg GAE/g of extract) [10]. By neutralizing DPPH, we achieved an 88% concentration of 50 μg/mL and the IC50 value of *B. cordata* extract was 21.92 μg/mL. In vivo superoxide dismutase (SOD) and glutathione peroxidase (GPx) activities in the kidney cortex were diminished in diabetes mellitus (DM) rats, which contrasted with the control normoglycemic rats; the administration of the methanol extract of leaves of *B. cordata* effectively prevented the decrease in SOD and GPx with a dose of 100 mg/kg (*p* < 0.05, respectively).

Table 2 shows the whole animal data for glycemia, total body weight, urinary volume, and uptake of food and water in different experimental groups. All STZ-induced diabetic rat groups showed significant hyperglycemia (DM, 500 ± 14 mg/dL vs. control 100 ± 3 mg/dL, *p* < 0.05). DM+ Vit E and DM + EMB at 100 mg/kg decreased hyperglycemia. DM rats ingested more water and excreted a greater volume of urine compared with normoglycemic rats (144 ± 9 vs. 34 ± 4 9 mL/24 h; 102 ± 4 vs. 23 ± 3 mL/24 h). Concerning the methanolic extract, only urinary excretion was reduced with a dose of 50 mg/kg (Appendix A in Appendix A).

Table 2 shows a comparison of glycemia, body weight, water and food ingestion, and urinary volume: control (c), untreated diabetes mellitus (DM), diabetes mellitus treated with vitamin E (Vit E 250 mg/kg), diabetes mellitus treated with captopril (25 mg/kg), and diabetes mellitus treated with methanol extract of *B. cordata* (50, and 100 mg/kg), (EMB). * *p* < 0.05 control vs. treatment and # *p* < 0.05 DM vs. treatment.

### 2.2. Renal Hypertrophy Was Determined Through the Renal Weight/Total Body Weight Proportion

In Figure 2, the relationship was higher for DM (4.21 ± 0.1 mg/g) than the control (3.13 ± 0.08 mg/g). On the other hand, groups that received treatments with captopril (4.34 ± 0.2 mg/g), vitamin E (4.37 ± 0.1 mg/g), EMB at 50 mg/kg (4.61 ± 0.1 mg/g), and EMB at 100 mg/kg (4.76 ± 0.3 mg/g) did not show a reduction in kidney weight, due to the fact that the kidney weight/body weight proportion was observed to be similar to DM (Figure 2).

### 2.3. Proteins and Creatinine Clearance

Figure 3a shows the results obtained by quantifying the concentration of proteins excreted in the urine, which increased significantly in DM (68.83 ± 4.7 mg/24 h) compared to the control group (34.43 ± 4.4 mg/24 h). This represents how 68% more proteins were excreted in diabetic rats compared to the control group. In the groups that received the treatments, a significant decrease in protein excretion was observed compared to DM. Another parameter that indicates kidney damage is a decrease in creatinine clearance. In this regard, Figure 3b shows that DM significantly decreased creatinine clearance, which was 0.36 ± 0.02 mL/min, in contrast to the normoglycemic control group (1.01 ± 0.08 mL/min). Creatinine clearance increased in the treated groups compared to DM. The results obtained were DM + Cap (0.62 ± 0.02 mL/min), DM + Vit E (0.52 ± 0.05 mL/min), and DM-EMB 100 mg/kg (0.52 ± 0.05 mL/min).

### 2.4. Effect of MEB on NFκB Translocation from Cytoplasm to Nucleus

The translocation of transcription factor NFκB from the cytoplasm to the nucleus of cortex renal cells increased significantly in DM rats (1.134 01 ± 0.1 u.a.) compared to the normoglycemic control group (0.614 ± 0.01 u.a.); this increase corresponds to 60%. The DM-MEB 100 mg/kg group significantly decreased the translocation of NFkB (0.291 ± 0.04 u.a.) compared to DM and the control (Figure 4a). The DM-Cap (0.6 ± 0.12 u.a.), DM-Vit E (0.622 ± 0.01 u.a.), and DM-EMB 50 mg/kg (0.704 ± 0.2 u.a.) groups decreased the nuclear translocation of this factor compared to DM rats.

In Figure 4b, the expression of NFκB in the cytoplasm DM group (0.29 ± 0.03 u.a.) decreased significantly by up to 64%, contrasting with the control normoglycemic rats (0.82 ± 0.1 u.a.). The groups treated with EMB (50 and 100 mg/kg) increased the cytoplasmic presence of NFkB (0.782 ± 0.1 and 0.745 ± 0.1 a.u.) compared to the DM group. They restored its expression, almost in its entirety, in contrast with the control normoglycemic rats. The rats administered captopril and vitamin E also exhibited increased cytoplasmic expression of NFκB (0.601 ± 0.05 a.u. and 0.481 ± 0.09 a.u., respectively) (Figure 4b).

### 2.5. Effect of EMBC on PPARγ Expression

The results obtained show that, in DM rats, the expression of transcription factor PPARγ significantly decreased (0.951 ± 0.1 u.a.) compared to the normoglycemic control group (1.49 ± 0.2 u.a.). The DM groups treated with EMB (50 and 100 mg/kg) showed a tendency to increase the expression of PPARγ (0.99 ± 0.07 and 0.98 ± 0.1 a.u., respectively). The DM groups treated with captopril and vitamin E showed similar results to the DM group for DM-Cap (0.884 ± 0.03 u.a) and DM-Vit E (0.942 ± 0.01 u.a) (Figure 5).

## 3. Discussion

The results obtained in the present work show that MEB has a nephroprotective effect by reducing proteinuria and increasing creatinine clearance. It decreases the translocation of NFκB from the cytoplasm to the nucleus, which indicates that the expression of cytokines that participate in the inflammatory process are decreased; these effects could therefore be due to the naringenin, routine, quercetin, apigenin, α-amyrin, and ferulic acid present in the methanolic extract of leaves of *B. cordata*.

Secondary metabolites present antioxidant activity by counteracting or eliminating free radicals generated in the body. As such, one of the techniques that is commonly used to evaluate the elimination of radicals is DPPH; the results of this study demonstrated the ability of *B. cordata* to eliminate DPPH radicals. The IC50 of MEB was 21.92 μg/mL, in comparison with quercetin (14.5 μg/mL) (standard) and other extracts, such as the aqueous extract of *Dianthus basuticus* (6.95 μg/mL) [10] and ethanolic extract of *Chiranthodendron pentadactylon* (18.05 µg/mL) [11]. These results indicate that methanol extract from the leaves of *B. cordata* presents renoprotective effects against kidney injury caused by DM, at least in part, through a reduction in oxidative stress; recent studies have shown that naringenin can reduce oxidative stress by stimulating transcription factors such as nuclear factor erythroid 2 (Nrf2) [12].

Various drugs have been used to induce diabetes mellitus, such as streptozotocin in pancreatic beta cells, which enters through the GLUT-2 transporter, preventing the passage of glucose into the cell β which generates cell damage and a deficit in the expression of proinsulin, generating a state of hyperglycemia and a decrease in body weight within the first 2 to 4 days after induction [13]. It has been reported that some secondary metabolites, such as saponins, steroids, and flavonoids, have hypoglycemic effects, like alpha amyrin and ferulic acid, which are present in the ethanolic extract of *B. cordata*; the administration of 100 mg/kg produced a hypoglycemic effect in a similar way to that found in other plants such as *Dainthus basuticus* [10] and *Bromelia karatas* [14]. 

Other characteristics observed in this model are polyuria, polydipsia, and weight loss. The presence of insulin regulates anorexigenic peptides’ control of satiety and appetite, because, in DM, as there is no insulin available, the regulation of these peptides is altered, which leads to an increase in food consumption [15]. For polyuria, excess glucose filtered by the kidneys is not reabsorbed, which generates water retention by osmosis in proximal convoluted tubules and, therefore, the body eliminates a greater volume of water and electrolytes, which leads to a greater intake of water [15]. The results obtained show that diabetic rats excreted a greater volume of urine compared to normoglycemic rats. Those DM rats treated with captopril, vitamin E, and EMB (50 mg/kg) significantly decreased the volume of urine excreted. However, the group treated with EMB (100 mg/kg) excreted a greater volume of urine. This result could be because some of the *B. cordata* compounds have diuretic properties, which produce greater urine excretion [16].

Several studies have shown that, at the beginning of DN, there is an increase in the weight of kidneys, as well as the area of proximal tubule cells; this structural alteration is known as renal hypertrophy and is associated with the overexpression of TGF-β and CTGF, especially in cells of the proximal convoluted tubules and mesangial cells [17,18]. DM rats treated with captopril and vitamin E presented a significant decrease in the group of diabetic rats. Captopril prevents the conversion of angiotensin I (Ang I) into pleiotropic peptide hormone angiotensin II (Ang II). Since there is no Ang II, the expression of transforming growth factor beta is reduced, which slows the progression of DN [19].

The effect of vitamin E (250 mg/kg) is attributed to its antioxidant capacity and modulation of the production of multiple polypeptide growth factors, for example, connective tissue growth factor (CTGF), TGF-β or IGF-1, which promote renal hypertrophy [20]. Regarding the methanolic extract of *B. cordata*, studies carried out in NRK-52E cell cultures showed that the administration of narigenin reduces cell proliferation and renal hypertrophy in a dose-dependent manner [20].

Proteinuria and renal clearance were determined to be indicators of kidney damage. According to the results obtained, hyperglycemia favors an increase in glomerular permeability and intraglomerular pressure, which facilitates proteinuria. Treatment with vitamin E decreased proteinuria compared to a group of diabetic rats. Various studies suggest that vitamin E can reduce the incidence of complications [21,22] due to its ability to capture free radicals and oxygen-reactive species. Treatments with EMB (50 and 100 mg/kg) decreased proteinuria; this result suggests that chemical compounds with an antioxidant capacity, such as those found in *B. cordata* [23] that are associated with secondary metabolite phenolic types like caffeic acid, ferulic acid, naringenin, chlorogenic, and apigenin acids, can restore the action speed of the antioxidant enzymes [21].

Chronic hyperglycemia induces excess reactive oxygen species production, decreases antioxidant capacities, and promotes the immune system to secrete inflammatory mediators and cytokines that harm glomerular capillaries and alter the renal tubular structure and function.

The expression of transcription factor NFκB was also determined both in the cytoplasm and nucleus of kidney cells; when analyzing these results, an increase in NFκB translocation from the cytoplasm and nucleus was observed in diabetic rats. These data correlate to those obtained by Huang et al. (2024) [22], who showed how NFκB activation was affected in diabetic rats and inhibited by multiple antioxidants such as ascorbic acid, Trolox, alpha-tocopherol, N-acetyl cysteine, β-carotene, and selenium.

In the groups treated with captopril and vitamin E, a greater expression of NFκB was observed in the cytoplasm and a lower expression in the nucleus. These data are comparable with those obtained by Di Vincenzo et al. (2019) [23], who administered antioxidants, including vitamin E, and observed that these compounds inhibited the activation of NFκB and decreased reactive oxygen species production but not hyperglycemia. The results suggest that, in diabetes mellitus, NFκB activation causes ND development and progression.

The expression of NFκB in groups treated with MEB is similar to that found in the control, which suggests that compounds present in MEB may be acting as antioxidants or inhibiting the activation of nuclear factor κB. Studies carried out in osteoblasts (cells responsible for bone growth) was observed that naringenin protects this type of cell against the oxidation and toxicity induced by oxidative stress through suppressing NFκB activation [24].

In the present work, the effect of EMB on the expression of PPARγ in the kidney cortex was also analyzed. PPARγ prevents the separation of NFκB from its inhibitor IκB and prevents its translocation to the nucleus and thus the transcription of proinflammatory cytokines [24].

The diabetic groups treated with captopril, vitamin E, and EMB 50 and 100 mg/kg also recorded a significant decrease compared to the control group. Some phenolic compounds present in plants act as natural ligands for PPARγ, regulating its activation [25]. Naringenin action on DN has been described and may be partially related to the induction of CYP4A-20-HETE and the up-regulation of PPARγ [25].

In terms of future perspectives, we could continue with the phytochemical analysis of the methanolic extract of *B. cordata* and isolate active compounds, chemically characterize them, study their mechanism of pharmacological action, and investigate whether the extract and its chemical fractions lack toxicity, which could lead to obtaining new drugs for the treatment of diabetic nephropathy. If required, the principles of medicinal chemistry could be applied to the structural optimization of new drugs, improving their pharmacological properties.

## 4. Materials and Methods

### 4.1. Elaboration and Analysis of the Methanol Extract of B. cordata

Leaves of *B. cordata* were collected in March 2016 at the Ecological Reserve, Pedregal de San Angel, Mexico. These were authenticated and conserved at the Herbarium of the Department of Botany of the FES Iztacala, UNAM (2524).

The extraction of the powder of leaves (30 g) was performed with Soxhlet equipment (IMPARLAB brand, Atizapán de Zaragoza, State of Mexico, Mexico, 50 mL, nozzles 55/50 and 24/40) using 300 mL of methanol at 55 °C. Residue was filtered and concentrated at 40 °C using a rotary evaporator (Buchii RE-111, Buchii, Meierseggstrasse 40, 9230 Flawil, Switzerland) coupled with a vacuum system (BuchiiVacV-153, Buchii, Meierseggstrasse 40, 9230 Flawil, Switzerland) and a cooling system (ECO20, Atlas Copco Group, NASDAQ OMX Stockholm, ATCO A, ATCO B, USA. This was then refrigerated at 4 °C in dark conditions until use. The extraction process was carried out to dryness under reduced pressure for the total elimination of alcohol. 

### 4.2. Herbal Extract’s Antioxidant Capacity

The quantification of total phenolic content (TPC) of the methanol extract of leaves of *B. cordata* (EMB) was realized by the modified Folin–Ciocalteu method, with gallic acid utilized as the standard phenolic compound. 2,2-diphenyl-1-picrylhydrazyl (DPPH) analysis was realized, as explained by Mirbagheri 2017 [26]. 

### 4.3. Phytochemical Analysis

For the chromatographic separation of SEM, a high-performance liquid chromatograph Hewlett Packard Mod. 1100 was employed, equipped with an automatic injector (Agilent Technologies Mod. 1200), a diodes array detector (Hewlett Packard Mod. 1100), and a quaternary pump HP Mod.1100. For the determination of flavonoids, phenolic acids, and terpenoids, the methodology described by [27] was used. Given the high polarity of the solvent, the methanolic extract of B. cordata is rich in various secondary metabolites; however, for this study, specific liquid chromatography conditions were used to detect phenolic acids and flavonoids on the one hand and terpenoids on the other. In the first case, the flavonoids were analyzed with a Hypersil ODS column (125 *×* 4 mm) from Hewlet Packard with a gradient of (a) water at pH 2.5 adjusted with trifluoroacetic acid and (b) acetonitrile with the following parameters: flow rate of 1 mL min^−1^ and temperature of 30 °C. The detector was adjusted to 254, 280, 330, and 365 nm. The HPLC standard library includes rutin, morin, quercetin, catechin, hesperidin, phloritzin, naringenin, phloretin, apigenin, myricetin, kaempferol, and isorhamnetin. For phenolic acids, a Nucleosil column (125 *×* 4 mm) from Macherey-Nagel was used, also with a gradient of (a) water at pH 2.5 adjusted with trifluoroacetic acid and (b) acetonitrile with the following parameters: flow rate of 1 mL min-1 and temperature of 30 °C. The detector was adjusted at 254, 280, and 330 nm. The library of standards that the HPLC has are protocatechuic acid, p-hydroxybenzoic acid, vanillic acid, caffeic acid, *β*-resorcylic acid, 3,5-dihydroxybenzoic acid, gallic acid, syringic acid, p-coumaric acid, chlorogenic acid, sinapic acid, ferulic acid, and rosmarinic acid. Finally, the conditions used for terpenoids were a Zorbax Eclipse XDB C-8 column (125 *×* 4 mm) from Agilent Technologies, isocratic analysis with (a) 80% acetonitrile and (b) 20% water, flow rate of 1 mL min^-1^, and temperature of 40 °C. The detector was adjusted to 220 nm. The HPLC standards library includes carnosol, ursolic acid, stigmasterol, oleanolic acid, *α*-amyrin, and *β*-sitosterol. 

### 4.4. Animals and Treatment

The Animal Use Ethics Committee of the FES Iztacala, UNAM revised the experimental investigation. It was authorized under Protocol No. CE/FESI/052017/1172. The use of laboratory animals was carried out under the indications for the care and use of laboratory animals of the Official Mexican Rule (NOM-062-ZOO-1999, updated in 2001), the International Guide for Caring and Use of Laboratory Animals NRC2002.

Male Wistar rats with a body weight of 280–300 g had free access to standard rat chow (Rodent Laboratory Chow 5001, Ralston Purina, St. Louis, MO, United States (USA) and sterilized tap water, with 12–12 h light–dark cycles. DM was provoked by a single streptozotocin (STZ) intraperitoneal (IP) administration (65 mg/kg of body weight in 10 mM sodium citrate buffer, pH 4.5). Normoglycemic control (C) rats were administered only vehicle solution (10 mM sodium citrate buffer, pH 4.5). Two days after STZ administration, glycemia was quantified in tail vein blood samples using a glucometer make (One Touch Accu Chek, Roche, Raleigh, NC, USA). Animals with glycemia > 300 mg/dL were included in the experimental study. DM rats were randomly distributed into (1) diabetic rats without treatment (DM) receiving vehicle (oil), (2) DM rats administered vitamin E, 250 mg/kg (DM + VIT E), (3) DM rats administered captopril (25 mg/kg), (4) (DM + CAP), and (5) and (6) diabetic rats administered with methanol extract of *Buddleja cordata* (50 and 100 mg/kg) (DM + EMB) [12]. Methanol extract of leaves of *B. cordata* was previously dissolved in propylene glycol (10%) in water for in vivo experiments, because previous studies have shown that the possible nephrotoxic effect of this solvent agent is achieved in rats via oral administration of concentrated solutions of 45%, at doses of 1000 mg/kg, for periods of 28 to 90 days [13,14].

The rats were located in metabolic cages to quantify food and water intake and urinary volume. Urine samples were immediately frozen and kept at −80 °C to measure proteins by the Bradford method and creatinine was measured with a commercial kit (Cayman Chemical, Ann Arbor, MI, USA).

### 4.5. Determination of Renal Hypertrophy

Kidney weight/total body weight proportion was employed as a kidney hypertrophy index. Total DNA and protein from cortex tissue were quantified by the Trizol reagent procedure (Invitrogen, USA) and the protein/DNA proportion was used as an indicator of relative hypertrophy.

### 4.6. Western Blotting

For the Western blot of proteins of NFkB p65 and PPARγ in the kidney cortex, we used the methodology described by Amato et al. (2016) [13].

### 4.7. In Vivo Antioxidant Activity

Catalase, superoxide dismutase, and glutathione peroxidase activities in renal cortical homogenates were measured by methods described in Guzman et al., 2023 [27]

### 4.8. Statistical Analysis

The statistical analyses were determined employing GraphPad Prism 5.0 software (GraphPad Software, Boston, MA, USA), treatments, and their interrelationships by one-factor analysis of variance (ANOVA). When the interrelationship and/or the main effects were considered statistically significant, means were compared employing Tukey’s post hoc test.

## 5. Conclusions

Methanolic extract from the leaves of *Buddleja cordata* has compounds that delay kidney damage, since it reduces proteinuria, increases creatinine clearance, and prevents the translocation of NFκB from the cytoplasm to the nucleus, preventing the production of proinflammatory mediators during diabetic nephropathy.

## Figures and Tables

**Figure 1 ijms-25-11432-f001:**
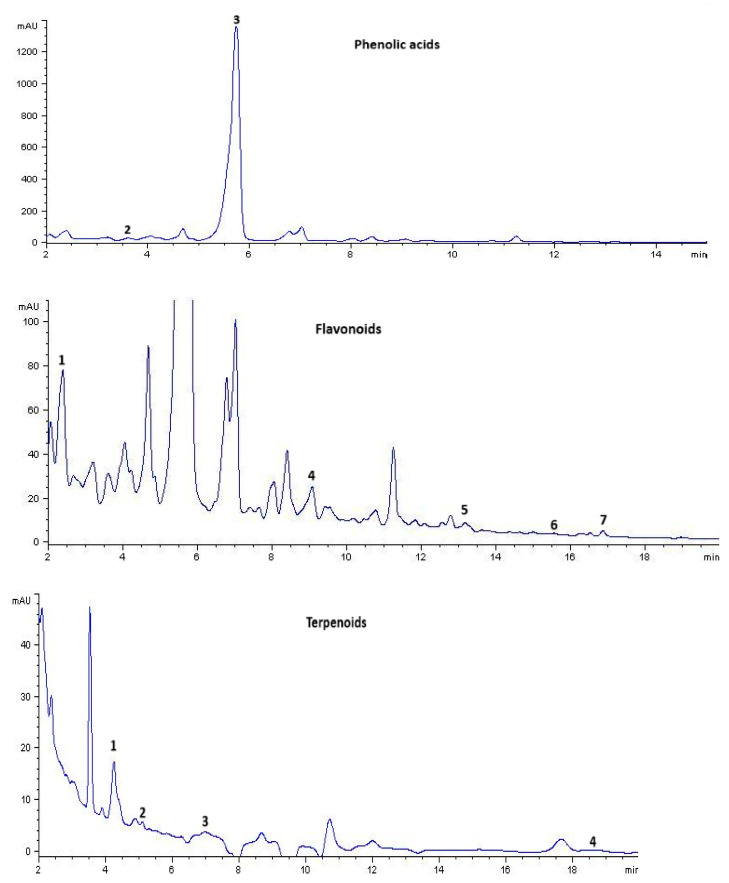
Chromatographic profile of terpenoids, phenolic acids, and flavonoids of methanolic extract of *B. cordata*.

**Figure 2 ijms-25-11432-f002:**
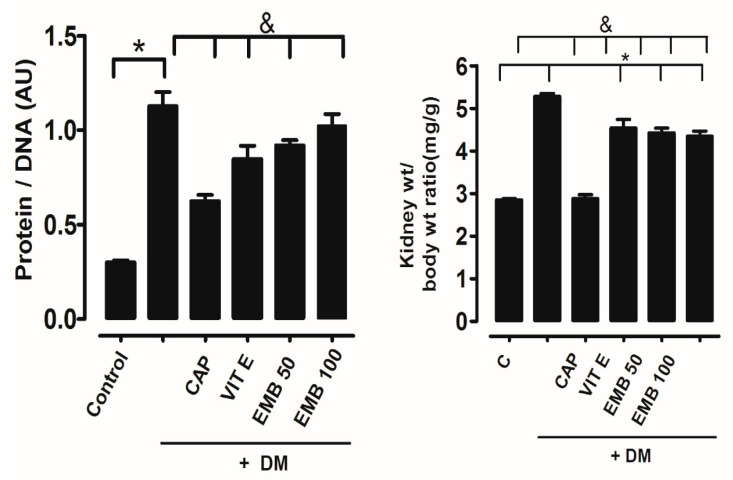
(a) Comparison of protein/DNA ratio (b) and kidney weight/body weight ratio. Control (c), untreated diabetes mellitus (DM), diabetes mellitus treated with vitamin E (Vit E 250 mg/kg), diabetes mellitus treated with captopril (25 mg/kg), and diabetes mellitus treated with methanol extract of *B. cordata* (50 and 100 mg/kg), (EMB). *p* < 0.05 * control vs. treatment; & DM vs. treatment.

**Figure 3 ijms-25-11432-f003:**
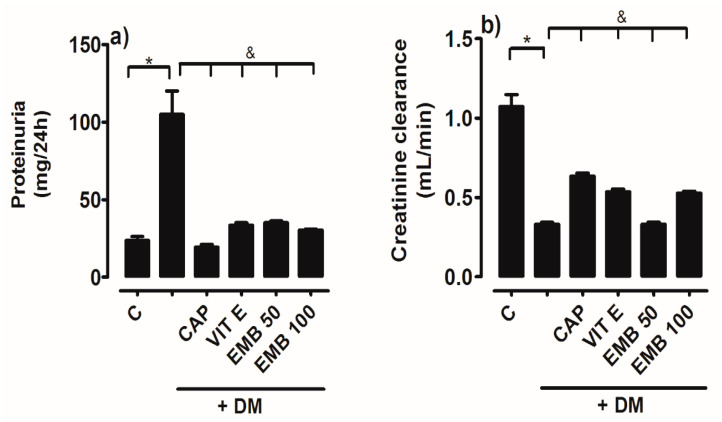
Comparison of proteinuria (**a**) and creatinine clearance (**b**) in groups of rats: control (C), untreated diabetes mellitus (DM), diabetes mellitus treated with vitamin E (DM + Vit E), diabetes mellitus treated with captopril (DM + CAP), and diabetes mellitus treated with methanol extract of B. cordata (DM + EMB). Data area expressed as mean ± SEM. * *p* < 0.05 Control vs. control vs. treatment; & DM vs. treatment.

**Figure 4 ijms-25-11432-f004:**
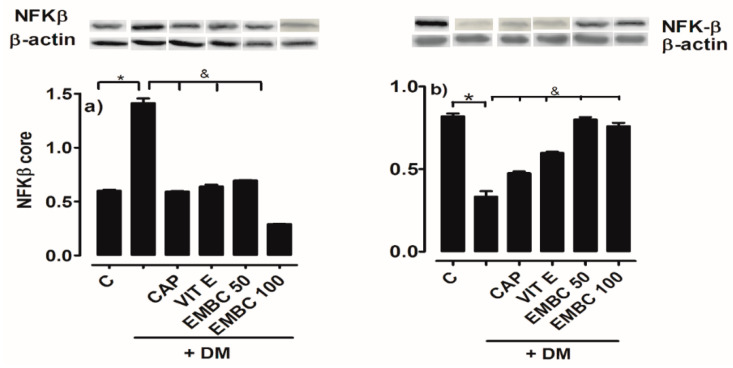
Effect of ethanol extract of B. cordata (EMB) on the expression of (**a**) NFkB core and (**b**) NFkB cytoplasm in renal cortex of diabetic rats (DM); diabetic + captopril (DM + CAP); diabetic + vitamin E (DM + Vit E); and diabetic + ethanol extract of B. cordata (DM + EMB). Values are expressed as mean ± SEM; n = 5; * *p* < 0.05 control vs. DM; & *p* < 0.05 DM vs. treatment.

**Figure 5 ijms-25-11432-f005:**
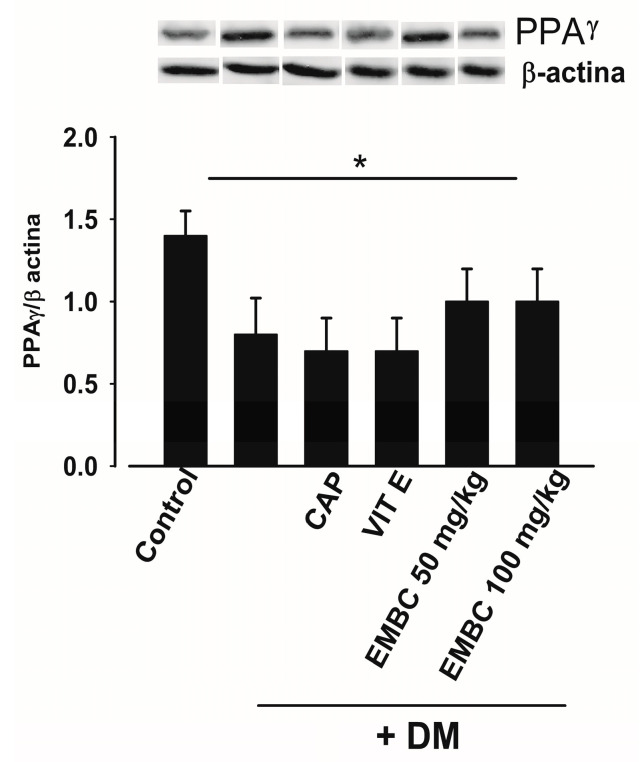
Effect of ethanol extract of *B. cordata* (EMB) on the expression of PPARγ in the renal cortex of diabetic rats (DM); diabetic + captopril (DM + CAP); diabetic + vitamin E (DM + Vit E); and diabetic + ethanol extract of *B. cordata* (DM + EMB). Values are expressed as mean ± SEM; *n* = 5; * = *p* < 0.05 control vs. DM.

**Table 1 ijms-25-11432-t001:** Phenolic compounds detected in the methanolic extract of *B. cordata*.

Peak No.	Tr (min)	Identification	Amount (mg Dry Extract g^−1^)
1	2.42	Catechin	3.4
2	3.60	Chlorogenic acid	0.66
3	5.72	Ferulic acid	56.74
4	9.05	Phloridzin	1.30
5	13.32	Quercetin	0.67
6	15.72	Naringenin	0.56
7	16.85	Apigenin	0.62

**Table 2 ijms-25-11432-t002:** Terpenoid compounds detected in the methanolic extract of *B. cordata*.

Peak No.	Tr (min)	Identification	Amount (μg Dry Extract g^−1^)
1	4.24	Oleanolic acid	12.69
2	5.04	Ursolic acid	0.69
3	7.02	α−αmyrin	223.22
4	18.71	β−sitosterol	1.63

## Data Availability

Data is contained within the article and Appendix A.

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
