# Peer review of "Effect of Buddleja cordata Leaf Extract on Diabetic Nephropathy in Rats"

_ijms, 2024, doi:10.3390/ijms252111432_

Round 1
Reviewer 1 Report
Comments and Suggestions for Authors
The article abstract clearly outlines the traditional use of Buddleja cordata and the research objectives. It focuses on its potential efficacy in the treatment of diabetic nephropathy. However, several aspects require major revision for clarity, accuracy and completeness. The following points should be addressed to improve the quality and impact of the paper:
Improve the introduction by highlighting recent research on the extraction and use of bioactive compounds from natural sources. Cite previous studies on Buddleja cordata extracts to underline the importance of this investigation. Provide an overview of Buddleja cordata, detailing its native habitat and geographical limitations. Is Buddleja cordata endemic to countries or regions?
The manuscript uses several abbreviations which should be defined the first time they appear in the text. For example, in line 49, EMP; line 62, DM; line 69, MEB. Make sure that all abbreviations are clearly defined.
Although it is not possible to calculate the uptake rates of Buddleja cordata extracts by rats, analysis of the metabolites present in their excreta after ingestion of the extracts may provide interesting data.
Table 1 should be revised to conform to journal guidelines. Although it is not necessary to show the peak areas of each component, the provision of quantitative data would enhance the informative value of the table. In addition, the definition of "%" in relation to Table 1 should be clarified.
Each component in Table 1 should correspond to the peaks in the chromatogram shown in Figure 1. For example, it appears that the peak for naringenin is not detected in the chromatogram. Ensure that all components are accurately represented.
The chromatogram in Figure 1 needs a major revision. The methodology for separation and analysis of the three types of compounds from B. cordata extracts should be clearly explained. All retention times should be standardized. The peak at the 5 min mark for flavonoids currently crosses the y-axis and should be corrected.
The manuscript states on line 59 that there is a high phenolic content. This statement should be supported by reference to other studies comparing the phenolic content of Buddleja cordata with that of other plants.
The use of methanol as an extraction solvent should be justified. Typically, a mixture of chloroform and methanol or methanol alone is used to extract a wide range of constituents. This should be explained in the context of the study.
The percentages in Table 1 do not add up to 100 %. What constituents other than phenolic acids, flavonoids and terpenoids were extracted from the plant? This should be addressed.
The manuscript should include detailed information on the HPLC (high performance liquid chromatography) experimental set-up. This includes the detector, wavelength, column type and other relevant conditions used in the measurements.
Author Response
To facilitate the visualization of the changes made to the manuscript, they were underlined in green.
Revisor 1
The methodology for the separation and analysis of the three types of compounds from B. cordata extracts should be clearly explained. All retention times should be standardized.
The manuscript should include detailed information about the HPLC (high performance liquid chromatography) experimental setup. This includes the detector, wavelength, column type, and other relevant conditions used in the measurements.
In the document from line 257 to 273 the methodology used for the identification of secondary metabolites is detailed by chromatographic analysis
The manuscript indicates on line 59 that there is a high phenolic content. This statement should be supported by references to other studies comparing the phenolic content of Buddleja cordata with that of other plants.
The manuscript indicates on lines 60 and 61 that the comparison of the in vitro antioxidant effect in relation to Hipocratea excelsa and Chiranthodendron pentadactylon Larreat was made and according to the research carried out by Guzman in 2020 and Santiago-Balmaseda in 2023, they showed antioxidant effects similar to those found in the present study.
The manuscript uses several abbreviations that should be defined the first time they appear in the text. For example, on line 49, EMP; line 62, DM; line 69, SEM. Please ensure that all abbreviations are clearly defined. Although it is not possible to calculate the absorption rates of Buddleja cordata extracts by rats, analysis of the metabolites present in their excrement after ingestion of the extracts may provide interesting data.
We do not agree with the reviewer's comment regarding the quantification of the absorption of secondary metabolites in animal excrement because to quantify this parameter it must be done in plasma, however, the objective of this study was to show that the methanolic extract of B. cordata leaves is effective for diabetic nephropathy, so it is not ruled out.
The use of methanol as an extraction solvent must be justified. Typically, a mixture of chloroform and methanol or methanol alone is used to extract a wide range of components.
For the extraction of secondary metabolites with polar characteristics, methanol is normally used as a solvent.
Table 1 should be revised to comply with the journal guidelines. Although peak areas for each component do not need to be shown, the provision of quantitative data would improve the informative value of the table. In addition, the definition of "%" in relation to Table 1 should be clarified. Each component in Table 1 should correspond to the peaks in the chromatogram shown in Figure 1. For example, it appears that the naringenin peak is not detected in the chromatogram. Please ensure that all components are accurately represented. The chromatogram in Figure 1 needs a major revision. The methodology for the separation and analysis of the three types of compounds from B. cordata extracts should be clearly explained. All retention times should be standardized.
Figure 1 and Table 1 were reviewed again, and the corresponding modifications were made according to the comments made by reviewers 1 and 2, and the corresponding modifications were made to the manuscript from lines 60-87.
The peak at the 5-minute mark for flavonoids currently crosses the y-axis and should correct.
In this case, it was decided to leave this scale so that the reader could appreciate the presence of the flavonoids present in the sample, since if the scale is adjusted to observe the full peak at time 5, the resolution of the others is lost.
The methodology for the detection of secondary metabolites was modified according to the observations made by reviewers 1 and 2. These modifications are found in rows 294-318.
The percentages in Table 1 do not add up to 100%. What components besides phenolic acids, flavonoids and terpenoids were extracted from the plant? This should be addressed.
The study on the components of this species focused basically on phenolic compounds (phenolic acids and flavonoids) as well as terpenoids, so the sum of these would not give 100% of the plant's components. For this reason and at your suggestion, it was decided to report the quantity in mg that corresponds to each of the metabolites found. On the other hand, it should be noted that the library of standards available on the chromatography equipment is 14 phenolic acids and 12 flavonoids.
To facilitate the visualization of the changes made to the manuscript, they were underlined in green.
Revisor 2
Some modifications were made to the introduction and a paragraph on future perspectives was added based on the reviewers' comments.
In the document, from lines 317 to 326, the contribution of each co-author to the work is detailed.
References have been modified according to journal guidelines
The use of a methanol solution to treat animals is generally not considered justified. Methanol is toxic to many organisms, including animals, and can cause serious health problems such as central nervous system depression, metabolic acidosis, and even death. If methanol is used as a solvent in research or treatment, its safety must be carefully evaluated and it must be completely removed or neutralized before being administered to animals. Other, less toxic solvents (ethanol, DMSO, glycerol, vegetable oils) are generally preferred for veterinary purposes.
The leaf powder was extracted using a Soxhlet apparatus, the residue was filtered and concentrated at 40 °C using a rotary evaporator coupled to a vacuum system and refrigerated at 4 °C in dark conditions until use. The extraction process was carried out at dryness under reduced pressure for the total elimination of alcohol, for administration to animals it was carried out with propylene glycol (10%) in water, for in vivo experiment, because previous studies have shown that the possible nephrotoxic effect of this solvent agent is achieved in rats via oral administration of concentrated solutions of 45%, at doses of 1000 mg/kg, for periods of 28 to 90 days.
In manuscript corresponding clarification was made in lines 239-246 and 290-293.
Table 2 seems unreliable and needs further explanation. Why, for example, were no statistical differences found/measured between the DM, Vit E, CAP, EMB 50 and EMB 100 groups and the control in the glycaemic parameter, when it is clear that the value ranges do not overlap? Did the application of vitamin A have a statistical effect on reducing glycaemia? Why were no statistical differences indicated for the water intake parameter compared to the control? To demonstrate the reliability of the measurements, it would be good to show the parameters obtained for all individuals evaluated as a supplementary file. Also, please improve the layout and clarity of the Tables and Figures.
According to the remarks made by reviewer 2 corresponding to table 2, the manuscript showed significant differences between the different groups, which were indicated in the table, and also in the description of the table the corresponding remarks were made in relation to the abbreviations.
My concern is with the setup of the experiment, i.e. the use of a methanol solution to treat animals (see objectives lines 45-47) or the comparison of individuals of different body weights consuming different volumes of water (Table 2).
For the extraction of secondary metabolites, the extract was extracted with methanol, concentrated and left to dry for 24 hours to eliminate the presence of methanol. For administration to animals, the extract was resuspended in propylene glycol. In relation to the diabetes mellitus model, the most relevant parameters of this disease are shown, such as polyuria, polydipsia, polyphagia and weight loss. Normally, in diabetic animals there is a noticeable weight loss, increased food intake, water and urinary volume are characteristics to be found in this model, so in the presence of the different treatments it would be expected that they would be modified with respect to only diabetic rats without treatment.
What do the values ​​below zero mean in the chromatogram in Figure 1, top panel for retention times of approximately 8 minutes? What happened to the baseline in the bottom panel?
Sometimes, a compound may present a lower absorbance than even the mobile phase and it is at that moment that a negative peak may appear, although that same compound when detected at a wavelength where it has its maximum absorption, emits the appropriate signal to be able to observe a positive peak

Reviewer 2 Report
Comments and Suggestions for Authors
Comments on the manuscript “Effect of leaves extract of Buddleja cordata on diabetic nephropathy in rats” by Elizabeth Alejandrina Guzmán Hernández et al., submitted to IJMS mdpi special issue Chronic Kidney Disease: The State of the Art and Future Perspectives.
Diabetic nephropathy is a severe complication of diabetes mellitus that involves kidney damage and can progress to chronic kidney disease and eventually kidney failure. This condition is particularly concerning as it impacts the quality of life of diabetic patients and raises their risk of illness and death. The study emphasizes the need for effective treatments, such as natural extracts like Buddleja cordata, to help manage and potentially reduce the impact of this complication.
The study found that the methanolic extract from the leaves of Buddleja cordata has compounds that can delay kidney damage in diabetic nephropathy. Specifically, it was observed that the extract reduces proteinuria, increases creatinine clearance, and prevents the translocation of NFκB from the cytoplasm to the nucleus, thereby avoiding the production of pro-inflammatory mediators during diabetic nephropathy
The article seems quite interesting but is not suitable for publication in its current form.
My concern is with the set-up of the experiment, i.e. using a methanol solution for treating animals (see objectives from lines 45-47) or comparing individuals of different body weights who consume different volumes of water (Table 2). Using a methanol solution for treating animals is generally not considered justified. Methanol is toxic to many organisms, including animals, and can cause serious health issues such as central nervous system depression, metabolic acidosis, and even death. If methanol is used as a solvent in research or treatment, it must be carefully evaluated for safety and thoroughly removed or neutralized before being administered to animals. Other, less toxic solvents (ethanol, DMSO, glycerol, vegetable oils) are usually preferred for veterinary purposes. Please explain whether a methanol extract was really used or it was just for HPLC analysis. If it was used please justify.
What do the values below zero in the chromatogram of Fig. 1, upper panel for retention times of approx. 8 minutes mean? What happened to the baseline in lower panel?
Table 2 seems to be unreliable and needs more explanation. Why, for example, no statistical differences were found/measured between the DM, Vit E, CAP, EMB 50, and EMB 100 groups and the control in the glycaemic parameter, when it is clear that the ranges of values do not overlap? Did the Vit A application have a statistical effect on lowering glycemia? Why were no statistical differences indicated for the water ingestion parameter compared to the control? To prove the reliability of the measurements, it would be good to show the parameters obtained for all tested individuals as a supplementary file.
Additionally, improve the layout and clarity of Tables and figures.
Discussion does not show the future perspectives which should be the clue of the paper submitted to the specific issue Chronic Kidney Disease: The State of the Art and Future Perspectives.
Expand the Conclusions chapter, I still not understand why methanolic extract was used
Technical issues:
Acronyms/Abbreviations/Initialisms should be defined the first time they appear in each of three sections: the abstract; the main text; the first figure or table. When defined for the first time, the acronym/abbreviation/initialism should be added in parentheses after the written-out form.
Materials and methods: Please give all the details concerning material and equipment name, type, producer, city and country.
The language of the manuscript is difficult to read. Please ask a native speaker for help.
References are not formatted according to the requirements of IJMS mdpi journal.
Author contributions: For research articles with several authors, a short paragraph specifying their individual contributions must be provided. The following statements should be used "Conceptualization, X.X. and Y.Y.; Methodology, X.X.; Software, X.X.; Validation, X.X., Y.Y. and Z.Z.; Formal Analysis, X.X.; Investigation, X.X.; Resources, X.X.; Data Curation, X.X.; Writing – Original Draft Preparation, X.X.; Writing – Review & Editing, X.X.; Visualization, X.X.; Supervision, X.X.; Project Administration, X.X.; Funding Acquisition, Y.Y.”, please turn to the CRediT taxonomy for the term explanation.
Comments on the Quality of English LanguageThe English used in the manuscript is sometimes difficult to understand. The sentences contain many stylistic and grammatical errors. They are sometimes too long and have no verbs.
For example,I give lines 45-47 containing the Objectives of the study which is difficoult to understand.
Author Response
Revisor 2
Some modifications were made to the introduction and a paragraph on future perspectives was added based on the reviewers' comments.
In the document, from lines 317 to 326, the contribution of each co-author to the work is detailed.
References have been modified according to journal guidelines
The use of a methanol solution to treat animals is generally not considered justified. Methanol is toxic to many organisms, including animals, and can cause serious health problems such as central nervous system depression, metabolic acidosis, and even death. If methanol is used as a solvent in research or treatment, its safety must be carefully evaluated and it must be completely removed or neutralized before being administered to animals. Other, less toxic solvents (ethanol, DMSO, glycerol, vegetable oils) are generally preferred for veterinary purposes.
The leaf powder was extracted using a Soxhlet apparatus, the residue was filtered and concentrated at 40 °C using a rotary evaporator coupled to a vacuum system and refrigerated at 4 °C in dark conditions until use. The extraction process was carried out at dryness under reduced pressure for the total elimination of alcohol, for administration to animals it was carried out with propylene glycol (10%) in water, for in vivo experiment, because previous studies have shown that the possible nephrotoxic effect of this solvent agent is achieved in rats via oral administration of concentrated solutions of 45%, at doses of 1000 mg/kg, for periods of 28 to 90 days.
In manuscript corresponding clarification was made in lines 239-246 and 290-293.
Table 2 seems unreliable and needs further explanation. Why, for example, were no statistical differences found/measured between the DM, Vit E, CAP, EMB 50 and EMB 100 groups and the control in the glycaemic parameter, when it is clear that the value ranges do not overlap? Did the application of vitamin A have a statistical effect on reducing glycaemia? Why were no statistical differences indicated for the water intake parameter compared to the control? To demonstrate the reliability of the measurements, it would be good to show the parameters obtained for all individuals evaluated as a supplementary file. Also, please improve the layout and clarity of the Tables and Figures.
According to the remarks made by reviewer 2 corresponding to table 2, the manuscript showed significant differences between the different groups, which were indicated in the table, and also in the description of the table the corresponding remarks were made in relation to the abbreviations.
My concern is with the setup of the experiment, i.e. the use of a methanol solution to treat animals (see objectives lines 45-47) or the comparison of individuals of different body wei

Round 2
Reviewer 2 Report
Comments and Suggestions for Authors
Comments on the manuscript “Effect of leaves extract of Buddleja cordata on diabetic nephropathy in rats” by Elizabeth Alejandrina Guzmán Hernández et al., submitted to IJMS mdpi special issue Chronic Kidney Disease: The State of the Art and Future Perspectives.
Second review.
Diabetic nephropathy is a severe complication of diabetes that can lead to chronic kidney disease and kidney failure. The study highlights the potential of natural treatments, such as Buddleja cordata extract, in managing this condition. The methanolic extract from its leaves reduced proteinuria, increased creatinine clearance, and prevented the activation of pro-inflammatory mediators, helping delay kidney damage.
As I wrote before the article seems quite interesting but is still unsuitable for publication in its current form. Now the main issue is language. The Authors were asked to seek a native speaker's help which they didn’t. The first example is a sentence from lines 20-21:
“Objectives: Was examined the effect of methanol extract of leaves of B. cordata on the expression 20 of pro-inflammatory cytokines and antioxidant effect in diabetic nephropathy.”
My main concern was the set-up of the experiment, i.e. using a methanol solution for treating animals (see objectives from lines 45-47) or comparing individuals of different body weights who consume different volumes of water (Table 2). The first issue has been clarified in lines 353-357 that the methanol extract of leaves of B. cordata was dissolved previously in propylene glycol (10%) in water.
The second issue somehow …..wanished. The Authors simply deleted table which contained unreliable values and needed more explanation. Instead, in lines 108—114 states that “Table 2 shows whole animal data of glycemia, total body weight, urinary volume, 108 and uptake of food and water in different experimental groups. All STZ-induced diabetic 109 rat groups showed significant hyperglycemia (DM, 500 ± 14 mg /dL vs control 100 ± 3mg/dL, p < 0.05). In DM+ Vit E and DM + EMB 100 mg/kg decreased hyperglycemia. DM rats ingested more water and excreted a greater volume of urine compared with normo-glycemic rats (144 ± 9 vs 34 ± 4 9 mL/24h; 102 ± 4 vs 23 ± 3 mL/24h), concerning methanolic extract only urinary excretion was reduced with the dose of 50 mg/kg.
Table 2. Comparison glycemia, body weight, water and food ingestion, and urinary 116 volume: control (c), untreated diabetes mellitus (DM), diabetes mellitus treated with vit-117 amin E (Vit E 250 mg/kg), diabetes mellitus treated with captopril (25 mg/kg) and diabetes 118 mellitus treated with methanol extract of B. cordata (50, and 100 mg/kg), (EMB). *p <0.05 119 Control vs treatment, and # p<0.05 DM vs treatment.” While in the manuscript Table 2 shows Terpenoids compounds detected in the methanolic extract of B. cordata (see line 86). Please clarify this issues.
Did the Vit A application have a statistical effect on lowering glycemia? Why were no statistical differences indicated for the water ingestion parameter compared to the control? To prove the reliability of the measurements, it would be good to show the parameters obtained for all tested individuals as a supplementary file.
Technical issues:
Acronyms/Abbreviations/Initialisms should be defined the first time they appear in each of three sections: the abstract; the main text; the first figure or table. When defined for the first time, the acronym/abbreviation/initialism should be added in parentheses after the written-out form.
Line 27 remove numbers after keywords
Lines 384-392 Author contributions, are still not prepared according to requrements of IJMS: For research articles with several authors, a short paragraph specifying their individual contributions must be provided. The following statements should be used "Conceptualization, X.X. and Y.Y.; Methodology, X.X.; Software, X.X.; Validation, X.X., Y.Y. and Z.Z.; Formal Analysis, X.X.; Investigation, X.X.; Resources, X.X.; Data Curation, X.X.; Writing – Original Draft Preparation, X.X.; Writing – Review & Editing, X.X.; Visualization, X.X.; Supervision, X.X.; Project Administration, X.X.; Funding Acquisition, Y.Y.”, please turn to the CRediT taxonomy for the term explanation.
Comments on the Quality of English LanguageThe manuscript still contains stylistic and grammatical errors. Please consult a native speaker for help.
Author Response
To facilitate the visualization of the changes made to the manuscript, they were underlined in green.
Revisor 1
The methodology for the separation and analysis of the three types of compounds from B. cordata extracts should be clearly explained. All retention times should be standardized.
The manuscript should include detailed information about the HPLC (high performance liquid chromatography) experimental setup. This includes the detector, wavelength, column type, and other relevant conditions used in the measurements.
Answer: In the document from line 257 to 273 the methodology used for the identification of secondary metabolites is detailed by chromatographic analysis.
For chromatographic separation of SEM a high-performance liquid chromatograph Hewlett Packard Mod. 1100 was employed, equipped with an automatic injector (Agilent Technologies Mod. 1200), equipped with a diodes array detector (Hewlett Packard Mod. 1100) and quaternary pump HP Mod.1100. For the determination of flavonoids, phenolic acids, and terpenoids, the methodology described by (15) was used. The methanolic extract of B. cordata is rich in various secondary metabolites given the high polarity of the solvent, however, for the purposes of this study, very specific liquid chromatography conditions were used to detect phenolic acids and flavonoids on the one hand and terpenoids on the other. In the first case, the flavonoids were analyzed with a Hypersil ODS column (125 x 4 mm) Hewlet Packard with a gradient of (a) water at pH 2.5 adjusted with trifluoroacetic acid and (b) acetonitrile with the following parameters: flow rate of 1 mL min-1, temperature of 30 oC, the detector was adjusted to 254, 280, 330 and 365 nm. The HPLC standard library includes: rutin, morin, quercetin, catechin, hesperidin, phloritzin, naringenin, phloretin, apigenin, myricetin, kaempferol and isorhamnetin. For phenolic acids a Nucleosil column (125x4 mm) Macherey-Nagel also with a gradient of (a) water at pH 2.5 adjusted with trifluoroacetic acid and (b) acetonitrile with the following parameters: flow rate of 1 mL min-1, temperature of 30 oC, the detector was adjusted at 254, 280 and 330 nm. The library of standards that the HPLC has are: protocatechuic acid, p-hydroxybenzoic acid, vanillic acid, caffeic acid, b-resorcium acid, 3,5-dihydroxybenzoic acid, gallic acid, syringic acid, p-coumaric acid, chlorogenic acid, sinapic acid, ferulic acid and rosmarinic acid. Finally, the conditions used for terpenoids were a Zorbax Eclipse XDB C-8 column (125 x 4 mm) Agilent Technologies, isocratic analysis with a) 80% acetonitrile and b) 20% water, flow rate of 1 mL min-1, temperature of 40oC and the detector was adjusted to 220 nm. The HPLC standards library is: carnosol, ursolic acid, stigmasterol, oleanolic acid, a-amyrin and b-sitosterol.
Answer: The manuscript indicates on line 59 that there is a high phenolic content. This statement should be supported by references to other studies comparing the phenolic content of Buddleja cordata with that of other plants.
The manuscript indicates on lines 60 and 61 that the comparison of the in vitro antioxidant effect concerning Chiranthodendron pentadactylon Larreat was made and according to the research carried out by Santiago-Balmaseda in 2023, they showed antioxidant effects similar to those found in the present study.
The manuscript uses several abbreviations that should be defined the first time they appear in the text. For example, on line 49, EMP; line 62, DM; line 69, SEM. Please ensure that all abbreviations are clearly defined. Although it is not possible to calculate the absorption rates of Buddleja cordata extracts by rats, analysis of the metabolites present in their excrement after ingestion of the extracts may provide interesting data.
Answer: We do not agree with the reviewer's comment regarding the quantification of the absorption of secondary metabolites in animal excrement because to quantify this parameter it must be done in plasma, however, the objective of this study was to show that the methanolic extract of B. cordata leaves is effective for diabetic nephropathy, so it is not ruled out.
The use of methanol as an extraction solvent must be justified. Typically, a mixture of chloroform and methanol or methanol alone is used to extract a wide range of components.
Answer: For the extraction of secondary metabolites with polar characteristics, methanol is normally used as a solvent.
Table 1 should be revised to comply with the journal guidelines. Although peak areas for each component do not need to be shown, the provision of quantitative data would improve the informative value of the table. In addition, the definition of "%" in relation to Table 1 should be clarified. Each component in Table 1 should correspond to the peaks in the chromatogram shown in Figure 1. For example, it appears that the naringenin peak is not detected in the chromatogram. Please ensure that all components are accurately represented. The chromatogram in Figure 1 needs a major revision. The methodology for the separation and analysis of the three types of compounds from B. cordata extracts should be clearly explained. All retention times should be standardized.
Figure 1 and Table 1 were reviewed again, and the corresponding modifications were made according to the comments made by reviewers 1 and 2, and the corresponding modifications were made to the manuscript from lines 60-87.
The peak at the 5-minute mark for flavonoids currently crosses the y-axis and should correct.
In this case, it was decided to leave this scale so that the reader could appreciate the presence of the flavonoids present in the sample, since if the scale is adjusted to observe the full peak at time 5, the resolution of the others is lost.
The redaction of the methodology for the detection of secondary metabolites by HPLC was modified according to the observations made by reviewers 1 and 2. These modifications are found in rows 294-318.
The percentages in Table 1 do not add up to 100%. What components besides phenolic acids, flavonoids and terpenoids were extracted from the plant? This should be addressed.
Answer: The study on the components of this species focused basically on phenolic compounds (phenolic acids and flavonoids) as well as terpenoids, so the sum of these would not give 100% of the plant's components. For this reason and at your suggestion, it was decided to report the quantity in mg that corresponds to each of the metabolites found. On the other hand, it should be noted that the library of standards available on the chromatography equipment is 14 phenolic acids and 12 flavonoids.
Revisor 2
Some modifications were made to the introduction and a paragraph on future perspectives was added based on the reviewers' comments.
Answer: From future perspectives, we could continue with the phytochemical analysis of the methanolic extract of B. cordata and isolate active compounds, characterize them chemically and study their mechanism of pharmacological action, and investigate whether the extract and its chemical fractions lack toxicity, which could lead to the obtaining new drugs for the treatment of diabetic nephropathy. If required, the principles of medicinal chemistry could be applied to the structural optimization of new drugs, improving their pharmacological properties.
In the document, from lines 317 to 326, the contribution of each co-author to the work is detailed.
Author Contributions: David Segura Cobos structured the intellectual content, performed the literature search, and performed experimental procedures, performed the literature search, carried out experimental studies, and elaborated the manuscript. Adriana Miranda Ocaña, Omar Ortiz performed experimental studies, performed data acquisition. Elizabeth Alejandrina Guzmán Hernández analyzed the data, performed statistical analysis, and prepared the manuscript. María Eugenia Garín Aguilar designed the study, defined the intellectual content, realized the literature search, and realized experimental work. Rubén San Miguel Chávez performed experimental procedures, data acquisition, data analysis, statistical analysis, and elaboration of the manuscript. Martín Palomar Morales performed the literature search and performed experimental studies.
References have been modified according to journal guidelines
The use of a methanol solution to treat animals is generally not considered justified. Methanol is toxic to many organisms, including animals, and can cause serious health problems such as central nervous system depression, metabolic acidosis, and even death. If methanol is used as a solvent in research or treatment, its safety must be carefully evaluated and it must be completely removed or neutralized before being administered to animals. Other, less toxic solvents (ethanol, DMSO, glycerol, vegetable oils) are generally preferred for veterinary purposes.
The leaf powder was extracted using a Soxhlet apparatus, the residue was filtered and concentrated at 40 °C using a rotary evaporator coupled to a vacuum system and refrigerated at 4 °C in dark conditions until use. The extraction process was carried out at dryness under reduced pressure for the total elimination of alcohol, for administration to animals it was carried out with propylene glycol (10%) in water, for in vivo experiment, because previous studies have shown that the possible nephrotoxic effect of this solvent agent is achieved in rats via oral administration of concentrated solutions of 45%, at doses of 1000 mg/kg, for periods of 28 to 90 days.
In manuscript corresponding clarification was made in lines 239-246 and 290-293.
Table 2 seems unreliable and needs further explanation. Why, for example, were no statistical differences found/measured between the DM, Vit E, CAP, EMB 50 and EMB 100 groups and the control in the glycaemic parameter, when it is clear that the value ranges do not overlap? Did the application of vitamin A have a statistical effect on reducing glycaemia? Why were no statistical differences indicated for the water intake parameter compared to the control? To demonstrate the reliability of the measurements, it would be good to show the parameters obtained for all individuals evaluated as a supplementary file. Also, please improve the layout and clarity of the Tables and Figures.
According to the remarks made by reviewer 2 corresponding to table 2, the manuscript showed significant differences between the different groups, which were indicated in the table, and also in the description of the table the corresponding remarks were made in relation to the abbreviations.
My concern is with the setup of the experiment, i.e. the use of a methanol solution to treat animals (see objectives lines 45-47) or the comparison of individuals of different body weights consuming different volumes of water (Table 2).
For the extraction of secondary metabolites, extraction was realized with methanol, concentrated, and left to dry for 24 hours to eliminate the presence of methanol. For administration to animals, the dry extract was resuspended in propylene glycol. About the diabetes mellitus model, the most relevant parameters of this disease are shown, such as polyuria, polydipsia, polyphagia, and weight loss. Normally, in diabetic animals there is a noticeable weight loss, increased food intake, water, and urinary volume are characteristics to be found in this model, so in the presence of the different treatments it would be expected that they would be modified concerning only diabetic rats without treatment.
What do the values ​​below zero mean in the chromatogram in Figure 1, top panel for retention times of approximately 8 minutes? What happened to the baseline in the bottom panel?
Sometimes, a compound may present a lower absorbance than even the mobile phase and it is at that moment that a negative peak may appear, although that same compound when detected at a wavelength where it has its maximum absorption, emits the appropriate signal to be able to observe a positive peak.

Round 3
Reviewer 2 Report
Comments and Suggestions for Authors
The authors have responded to all the comments in previous reviews
A linguistic error remained in line 307
A linguistic error remained in line 307